# BUILDING VISION MODELS UPON HEAT CONDUCTION

## ABSTRACT

Visual representation models leveraging attention mechanisms are challenged by significant computational overhead, particularly when pursuing large receptive fields. In this study, we aim to mitigate this challenge by introducing the Heat Conduction Operator (HCO) built upon the physical heat conduction principle. HCO conceptualizes image patches as heat sources and models their correlations through adaptive thermal energy diffusion, enabling robust visual representations. HCO enjoys a computational complexity of $O(N^{1.5})$, as it can be implemented using discrete cosine transformation (DCT) operations. HCO is plug-and-play, combining with deep learning backbones produces visual representation models (termed vHeat) with global receptive fields. Experiments across vision tasks demonstrate that, beyond the stronger performance, vHeat achieves up to a $3\times$ throughput, 80% less GPU memory allocation and 35% fewer computational FLOPs compared to the Swin-Transformer.

## 1 INTRODUCTION

Convolutional Neural Networks (CNNs) (Krizhevsky et al., 2012; He et al., 2016) have been the cornerstone of visual representation since the advent of deep learning, exhibiting remarkable performance across vision tasks. However, the reliance on local receptive fields and fixed convolutional operators imposes constraints, particularly in capturing long-range and complex dependencies within images (Luo et al., 2016). These limitations have motivated significant interest in developing alternative visual representation models, including architectures based on ViTs (Dosovitskiy et al., 2021; Liu et al., 2021) and State Space Models (Zhu et al., 2024; Liu et al., 2024). Despite their effectiveness, these models continue to face challenges, including relatively high computational complexity and a lack of interpretability.

When addressing these limitations, we draw inspiration from the field of heat conduction (Widder, 1976), where *spatial locality* is crucial for the transfer of thermal energy due to the collision of neighboring particles. Notably, analogies can be drawn between the principles of heat conduction and the propagation of visual semantics within the spatial domain, as adjacent image regions in a certain scale tend to contain related information or share similar characteristics. Leveraging these connections, we introduce **vHeat**, a physics-inspired vision representation model that conceptualizes image patches as *heat sources* and models the calculation of their correlations as the diffusion of thermal energy.

To integrate the principle of heat conduction into deep networks, we first derive the general solution of heat conduction in 2D space and extend it to multiple dimensions, corresponding to various feature channels. Based on this general solution, we design the **Heat Conduction Operator (HCO)**, which simulates the propagation of visual semantics across image patches along multiple dimensions. Notably, we demonstrate that HCO can be approximated through 2D (inverse) discrete cosine transformation (DCT/IDCT), effectively reducing the computational complexity to $\mathcal{O}(N^{1.5})$, Fig. 1. This improvement boosts both training and testing efficiency due to the high parallelizability of DCT and IDCT operations. Furthermore, as each element in the frequency domain obtained by DCT incorporates information from all patches in the image space, vHeat can establish long-range feature dependencies and achieve global receptive fields. To enhance the representation adaptability of vHeat, we propose learnable frequency value embeddings (FVEs) to characterize the frequency information and predict the thermal diffusivity of visual heat conduction.

We develop a family of vHeat models (*i.e.*, vHeat-Tiny/Small/Base), and extensive experiments are conducted to demonstrate their effectiveness in diverse visual tasks. Compared to benchmark vision

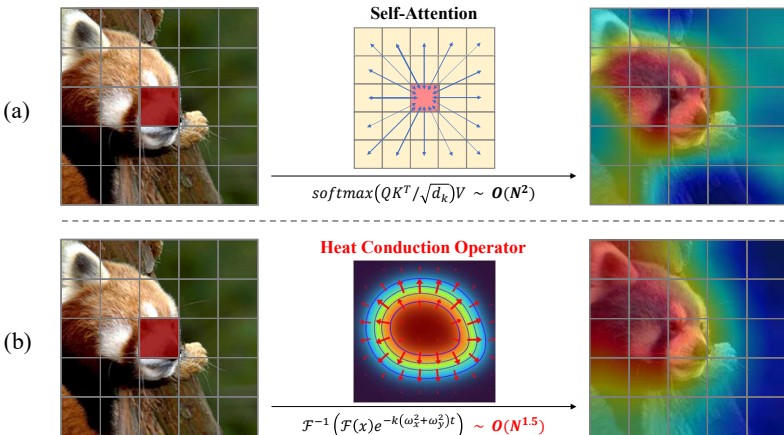

Figure 1: Comparison of information conduction mechanisms: self-attention *vs.* heat conduction. (a) The self-attention operator uniformly "conducts" information from a pixel to all other pixels, resulting in $\mathcal{O}(N^2)$ complexity. (b) The heat conduction operator (HCO) conceptualizes the center pixel as the heat source and conducts information propagation through DCT ($\mathcal{F}$) and IDCT ($\mathcal{F}^{-1}$), which enjoys interpretability, global receptive fields, and $\mathcal{O}(N^{1.5})$ complexity.

backbones with various architectures (*e.g.*, ConvNeXt (Liu et al., 2022b), Swin (Liu et al., 2021), and Vim (Zhu et al., 2024)), vHeat consistently achieves superior performance on image classification, object detection, and semantic segmentation across model scales. Specifically, vHeat-Base achieves a $84.0\%$ top-1 accuracy on ImageNet-1K, surpassing Swin by $0.5\%$, with a throughput exceeding that of Swin by a substantial margin over $40\%$ (661 *vs.* 456). To explore the generalization of vHeat, we've also validated its superiority on robustness evaluation benchmarks and low-level vision tasks. Besides, due to the $\mathcal{O}(N^{1.5})$ complexity of HCO, vHeat enjoys considerably lower computational cost compared to ViT-based models, demonstrating significantly reduced FLOPs and GPU memory requirements, and higher throughput as image resolution increases. In particular, when the input image resolution increases to $768 \times 768$, vHeat-Base achieves a $3\times$ throughput compared to Swin, with 80% less GPU memory allocation and 35% fewer computational FLOPs.

The contributions of this study are summarized as follows:

- We propose vHeat, a vision backbone model inspired by the physical principle of heat conduction, which simultaneously achieves global receptive fields, low computational complexity, and high interpretability.

- We design the Heat Conduction Operator (HCO), a physically plausible module conceptualizing image patches as heat sources, predicting adaptive thermal diffusivity by FVEs, and transferring information following the principles of heat conduction.

- Without bells and whistles, vHeat achieves promising performance in vision tasks including image classification, object detection, and semantic segmentation. It also enjoys higher inference speeds, reduced FLOPs, and lower GPU memory usage for high-resolution images.

## 2 RELATED WORK

**Convolution Neural Networks.** CNNs have been landmark models in the history of visual perception (LeCun et al., 1998; Krizhevsky et al., 2012). The distinctive characteristics of CNNs are encapsulated in the convolution kernels, which enjoy high computational efficiency given specifically designed GPUs. With the aid of powerful GPUs and large-scale datasets (Deng et al., 2009), increasingly deeper (Simonyan & Zisserman, 2014; Szegedy et al., 2015; He et al., 2016; Huang et al., 2017) and efficient models (Howard et al., 2017; Tan & Le, 2019; Yang et al., 2021; Radosavovic et al., 2020) have been proposed for higher performance across a spectrum of vision tasks. Numerous modifications have been made to the convolution operators to improve its capacity (Chollet, 2017), efficiency (Hua et al., 2018; Yu & Koltun, 2015) and adaptability (Dai et al., 2017; Wang et al.,

2023b). Nevertheless, the born limitation of local receptive fields remains. Recently developed large convolution kernels (Ding et al., 2022b) took a step towards large receptive fields, but experienced difficulty in handling high-resolution images.

**Vision Transformers.** Built upon the self-attention operator (Vaswani et al., 2017), ViTs have the born advantage of building global feature dependency. Based on the learning capacity of self-attention across all image patches, ViTs has been the most powerful vision model ever, given a large dataset for pre-training (Dosovitskiy et al., 2021; Touvron et al., 2021; Peng et al., 2022). The introduction of hierarchical architectures (Liu et al., 2021; Dong et al., 2022; Wang et al., 2021; Lu et al., 2021; Zhang et al., 2023; Tian et al., 2023; Dai et al., 2021; Ding et al., 2022a; Zhao et al., 2022) further improves the performance of ViTs. The Achilles' Heel of ViTs is the $\mathcal{O}(N^2)$ computational complexity, which implies substantial computational overhead given high-resolution images. Great efforts have been made to improve model efficiency by introducing window attention, linear attention and cross-covariance attention operators (Wang et al., 2020; Liu et al., 2021; Chen et al., 2021; Ali et al., 2021), at the cost of reducing receptive fields or non-linearity capacity. Other studies proposed hybrid networks by introducing convolution operations to ViTs (Wang et al., 2022; Dai et al., 2021; Vaswani et al., 2021), designing hybrid architectures to combine CNN with ViT modules (Dai et al., 2021; Srinivas et al., 2021; Lu et al., 2021).

**State Space Models and RNNs.** State space models (SSMs) (Gu et al., 2022; Nguyen et al., 2022; Wang et al., 2023a), which have the long-sequence modeling capacity with linear complexity, are also migrated from the natural language area (Mamba (Gu & Dao, 2023)). Visual SSMs were also designed by adapting the selective scan mechanism to 2-D images (Zhu et al., 2024; Liu et al., 2024). Nevertheless, SSMs based on the selective scan mechanism suffer from limited parallelism, restricting their overall potential. Recent receptance weighted key value (RWKV) and RetNet models (Peng et al., 2023; Sun et al., 2023) improved the parallelism while retaining the linear complexity. They combine the efficient parallelizable training of transformers with the efficient inference of RNNs, leveraging a linear attention mechanism and allowing formulation of the model as either a Transformer or an RNN, thus parallelizing computations during training and maintaining constant computational and memory complexity during inference. Despite the advantages, modeling a 2-D image as a sequence impairs interpretability.

**Biology and Physics Inspired Models.** Biology and physics principles have long been the fountain-head of creating vision models. Diffusion models (Song et al., 2020; Ho et al., 2020; Saharia et al., 2022), motivated by Nonequilibrium thermodynamics (De Groot & Mazur, 2013), are endowed with the ability to generate images by defining a Markov chain for the diffusion step. QB-Heat (Chen et al., 2022) utilizes physical heat equation as supervision signal for masked image modeling task. Spiking Neural Network (SNNs) (Ghosh-Dastidar & Adeli, 2009; Tavanaei et al., 2019; Lee et al., 2016) claims better simulation on the information transmission of biological neurons, formulating models for simple visual tasks (Bawane et al., 2018). The success of these models encourages us to explore the principle of physical heat conduction for the development of vision representation models.

## 3 METHODOLOGY

### 3.1 PRELIMINARIES: PHYSICAL HEAT CONDUCTION

Let $u(x, y, t)$ denote the temperature of point $(x, y)$ at time $t$ within a two-dimensional region $D \in \mathbb{R}^2$, the classic physical heat equation (Widder, 1976) can be formulated as

$$\frac{\partial u}{\partial t} = k \left( \frac{\partial^2 u}{\partial x^2} + \frac{\partial^2 u}{\partial y^2} \right), \tag{1}$$

where $k > 0$ is the **thermal diffusivity** (Bird, 2002), measuring the rate of heat transfer in a material. By setting the initial condition $u(x, y, t)|_{t=0}$ to $f(x, y)$, the general solution of Eq. equation 1 can be derived by applying the Fourier Transform (FT, denoted as $\mathcal{F}$) to both sides of the equation, which gives

$$\mathcal{F} \left( \frac{\partial u}{\partial t} \right) = k \mathcal{F} \left( \frac{\partial^2 u}{\partial x^2} + \frac{\partial^2 u}{\partial y^2} \right). \tag{2}$$

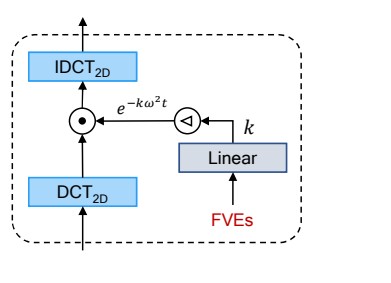
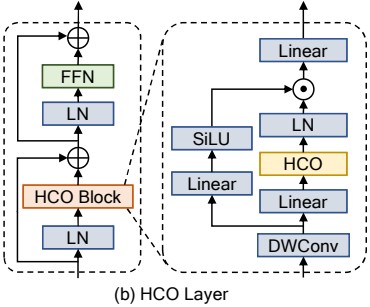

(a) Heat Conduction Operator        (b) HCO Layer

Figure 2: HCO and HCO layer. FVEs, FFN, LN, DWConv respectively denote frequency value embeddings, feed-forward network, layer normalization, and depth-wise convolution[1].

Denoting $\widetilde{u}(\omega_x, \omega_y, t)$ as the FT-transformed form of $u(x, y, t)$, *i.e.*, $\widetilde{u}(\omega_x, \omega_y, t) \coloneqq \mathcal{F}(u(x, y, t))$, the left-hand-side of Eq. equation 2 can be written as

$$\mathcal{F}\left(\frac{\partial u}{\partial t}\right) = \frac{\partial \widetilde{u}(\omega_x, \omega_y, t)}{\partial t}. \tag{3}$$

and by leveraging the derivative property of FT, the right-hand-side of Eq. equation 2 can be transformed as

$$\mathcal{F}\left(\frac{\partial^2 u}{\partial x^2} + \frac{\partial^2 u}{\partial y^2}\right) = -(\omega_x^2 + \omega_y^2)\widetilde{u}(\omega_x, \omega_y, t). \tag{4}$$

Therefore, by combining the expression of both sides of the equation, Eq. equation 2 can be formulated as an ordinary differential equation (ODE) in the frequency domain, which can be written as

$$\frac{d\widetilde{u}(\omega_x, \omega_y, t)}{dt} = -k(\omega_x^2 + \omega_y^2)\widetilde{u}(\omega_x, \omega_y, t). \tag{5}$$

By setting the initial condition $\widetilde{u}(\omega_x, \omega_y, t)|_{t=0}$ to $\widetilde{f}(\omega_x, \omega_y)$ ($\widetilde{f}(\omega_x, \omega_y)$ denotes the FT-transformed $f(x, y)$), $\widetilde{u}(\omega_x, \omega_y, t)$ in Eq equation 5 can be solved as

$$\widetilde{u}(\omega_x, \omega_y, t) = \widetilde{f}(\omega_x, \omega_y)e^{-k(\omega_x^2 + \omega_y^2)t}. \tag{6}$$

Finally, the general solution of heat equation in the spatial domain can be obtained by performing inverse Fourier Transformer ($\mathcal{F}^{-1}$) on Eq. equation 6, which gives the following expression

$$u(x, y, t) = \mathcal{F}^{-1}(\widetilde{f}(\omega_x, \omega_y)e^{-k(\omega_x^2 + \omega_y^2)t}) \tag{7}$$

$$= \frac{1}{4\pi^2}\int_{\widetilde{D}} \widetilde{f}(\omega_x, \omega_y)e^{-k(\omega_x^2 + \omega_y^2)t}e^{i(\omega_x x + \omega_y y)}d\omega_x d\omega_y. \tag{8}$$

## 3.2 vHeat: Visual Heat Conduction

Drawing inspiration from the analogies between the principles of physical heat conduction and the propagation of visual semantics within the spatial domain (*i.e.*, 'visual heat conduction'), we propose **vHeat**, a physics-inspired deep architecture for visual representation learning. The vHeat model is built upon the Heat Conduction Operator (HCO), which is designed to integrate the principle of heat conduction into handling the discrete feature of vision data. We also leverage the thermal diffusivity in the classic physical heat equation (Eq equation 1) to improve the adaptability of vHeat to vision data.

### 3.2.1 Heat Conduction Operator (HCO)

To extract visual features, we design HCO to implement the conduction of visual information across image patches in multiple dimensions, following the principle of physical heat conduction. To this end,

---

[1]Please refer to Sec. D.3 in Appendix, where we demonstrate that while depth-wise convolution aids in feature extraction, the primary improvements are attributed to the proposed HCO.

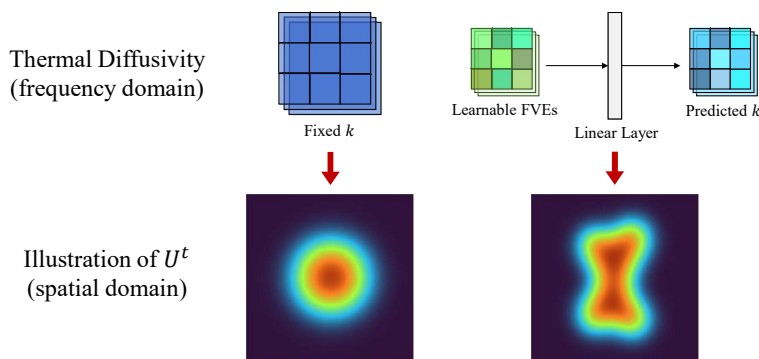

Figure 3: Illustration of temperature distribution $U^t$ *w.r.t.* thermal diffusivity $k$, given a heat source as the initial condition. The predicted $k$ leads to nonuniform visual heat conduction, which facilitates the adaptability of visual representation. (Best viewed in color)

we first extend the 2D temperature distribution $u(x, y, t)$ along the channel dimension and denote the resultant multi-channel image feature as $U(x, y, c, t)$ $(c = 1, \cdots, C)$. Mathematically, considering the input as $U(x, y, c, 0)$ and the output as $U(x, y, c, t)$, HCO simulates the general solution of physical heat conduction (Eq. equation 7) in visual data processing, which can be formulated as

$$U^t = \mathcal{F}^{-1}(\mathcal{F}(U^0)e^{-k(\omega_x^2 + \omega_y^2)t}), \tag{9}$$

where $U^t$ and $U^0$ are abbreviations for $U(x, y, c, t)$ and $U(x, y, c, 0)$, respectively.

For applying $\mathcal{F}(\cdot)$ and $\mathcal{F}^{-1}(\cdot)$ to discrete image patch features, it is necessary to utilize the discrete version of the (inverse) Fourier Transform (*i.e.*, DFT and IDFT). However, since vision data is spatially constrained and semantic information will not propagate beyond the border, we additionally introduce a common assumption of Neumann boundary condition (Cheng & Cheng, 2005), *i.e.*, $\partial u(x, y, t)/\partial \mathbf{n} = 0, \forall (x, y) \in \partial D, t \geq 0$, where $\mathbf{n}$ denotes the normal to the image boundary $\partial D$. As vision data is typically rectangular, this boundary condition enables us to replace the 2D DFT and IDFT with the 2D discrete cosine transformation, $\mathbf{DCT_{2D}}$, and the 2D inverse discrete cosine transformation, $\mathbf{IDCT_{2D}}$ (Strang, 1999). Therefore, the discrete implementation of HCO can be expressed as

$$U^t = \mathbf{IDCT_{2D}}(\mathbf{DCT_{2D}}(U^0)e^{-k(\omega_x^2 + \omega_y^2)t}), \tag{10}$$

and its internal structure is illustrated in Fig. 2(a). Particularly, the parameter $k$ stands for the thermal diffusivity in physical heat conduction and is predicted based on the features within the frequency domain (explained in the following subsection).

Notably, due to the computational efficiency of $\mathbf{DCT_{2D}}$, the overall complexity of HCO is $\mathcal{O}(N^{1.5})$, where $N$ denotes the number of input image patches. Please refer to Sec. B in Appendix for the detailed implementation of HCO using $\mathbf{DCT_{2D}}$ and $\mathbf{IDCT_{2D}}$.

### 3.2.2 ADAPTIVE THERMAL DIFFUSIVITY

In physical heat conduction, thermal diffusivity represents the rate of heat transfer within a material. While in visual heat conduction, we hypothesize that more representative image contents contain more energy, resulting in higher temperatures in the corresponding image features within $U(x, y, c, t)$. Therefore, it is suggested that the thermal diffusivity parameter $k$ should be learnable and adaptive to image content, which facilitates the adaptability of heat condution to visual representation learning.

Given that the output of $\mathbf{DCT}$ (i.e., $\mathbf{DCT_{2D}}(U^0)$ in Eq. equation 10) lies in the frequency domain, we also determine $k$ based on frequency values ($k := k(\omega_x, \omega_y)$). Since different positions in the frequency domain correspond to different frequency values, we propose to represent these values using learnable Frequency Value Embeddings (FVEs), which function similarly to the widely used absolute position embeddings in ViTsDosovitskiy et al. (2021) (despite in the frequency domain). As shown in Figure 2 (a), FVEs are fed to a linear layer to predict the thermal diffusivity $k$, allowing it to be non-uniform and adaptable to visual representations.

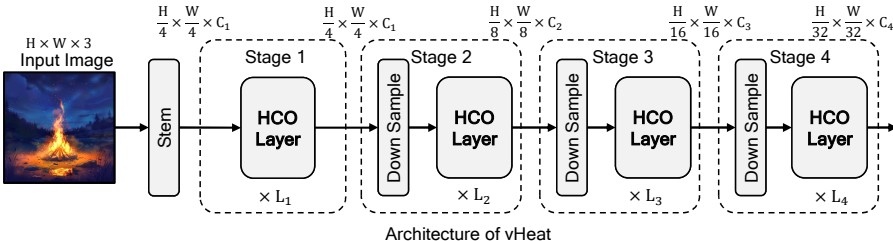

Figure 4: The network architecture of vHeat.

Practically, considering that $k$ and $t$ (the conduction time) are multiplied in Eq. equation 10, we empirically set a fixed value for $t$ and predict the values of $k$. Specifically, FVEs are shared within each network stage of vHeat to facilitate the convergence of the training process.

### 3.2.3 vHEAT MODEL

**Network Architecture.** We develop a vHeat model family including vHeat-Tiny (vHeat-T), vHeat-Small (vHeat-S), and vHeat-Base (vHeat-B). An overview of the network architecture of vHeat is illustrated in Fig. 4, and the detailed configurations are provided in Sec. C in Appendix. Given an input image with the spatial resolution of $H \times W$, vHeat first partitions it to image patches through a stem module, yielding a 2D feature map with $\frac{H}{4} \times \frac{W}{4}$ resolution. Subsequently, multiple stages are utilized to create hierarchical representations with gradually decreased resolutions of $\frac{H}{4} \times \frac{W}{4}$, $\frac{H}{8} \times \frac{W}{8}$, $\frac{H}{16} \times \frac{W}{16}$ and increasing channels. Each stage is composed of a down-sampling layer followed by multiple heat conduction layers (except for the first stage).

**Heat Conduction Layer.** The heat conduction layer, Fig. 2 (b), is similar to the ViTs block while replacing self-attention operators with HCOs and retaining the feed-forward network (FFN). It first utilizes a $3 \times 3$ depth-wise convolution layer. The depth-wise convolution is followed by two branches: one maps the input to HCO and the other computes the multiplicative gating signal like (Liu et al., 2024). HCO plays a crucial role in each heat conduction layer, Fig. 2 (b), where the mapped features from a linear layer are first processed by the $\mathbf{DCT_{2D}}$ operator to generate features in the frequency domain. Additionally, HCO takes FVEs as input for frequency representation to predict adaptive thermal diffusivity $k$ through a linear layer. By multiplying the coefficient matrix $e^{-k\omega^2 t}$ and performing $\mathbf{IDCT_{2D}}$, HCO implements the discrete solution of the visual heat equation, Eq. equation 10.

### 3.3 DISCUSSION

• **What is role of the thermal diffusivity coefficient** $e^{-k(\omega_x^2+\omega_y^2)t}$**?** When multiplying with $\mathbf{DCT_{2D}}(U^0)$, $e^{-k(\omega_x^2+\omega_y^2)t}$ acts as an adaptive filter in the frequency domain to perform visual heat conduction. Different frequency values correspond to distinct image patterns, $i.e.$, high frequency corresponds to edges and textures while low frequency corresponds to flat regions. With adaptive thermal diffusivity, HCO can enhance/depress these patterns within each feature channel. Aggregating the filtered features from all channels, vHeat achieves a robust feature representation.

• **Why does temperature** $U(x, y, c, t)$ **correspond to visual features?** Visual features are essentially the outcome of the feature extraction process, characterized by pixel propagation within the feature map. This process aligns with the properties of existing convolution, self-attention, and selective scan operators, exemplifying a form of information conduction. Similarly, visual heat conduction embodies this concept of information conduction through temperature, denoted as $U(x, y, c, t)$.

• **What is the relationship/difference between HCO and self-attention?** HCO dynamically propagates energy via heat conduction, enabling the perception of global information within the input image. This positions HCO as a distinctive form of attention mechanism. The distinction lies in its reliance on interpretable physical heat conduction, in contrast to self-attention, which is formulated through token similarity. Furthermore, HCO works in the frequency domain, implying its potential to affect all image patches through frequency filtering. Consequently, HCO exhibits greater efficiency

compared to self-attention, which necessitates computing the relevance of all pairs across image patches.

# 4 EXPERIMENT & ANALYSIS

Experiments are performed to assess vHeat and compare it against popular CNN and ViT models. Visualization analysis is presented to gain deeper insights into the mechanism of vHeat. The evaluation spans image classification, object detection, semantic segmentation, out-of-distribution classification, and low-level vision tasks. Please refer to Sec. C for experimental settings.

## 4.1 EXPERIMENTAL RESULTS

**Image classification.** The image classification results are summarized in Table 1. With similar FLOPs, vHeat-T achieves a top-1 accuracy of $82.2\%$, outperforming Swin-T by $0.9\%$, and Vim-S by $0.8\%$, respectively. Notably, the superiority of vHeat is also observed at both Small and Base scales. Specifically, vHeat-B achieves a top-1 accuracy of $84.0\%$ with only 11.2G FLOPs and 68M model parameters, outperforming Swin-B by $0.5\%$, and Vim-B by $0.8\%$, respectively.

In terms of computational efficiency, vHeat enjoys significantly higher inference speed across Tiny/Small/Base model scales compared to benchmark models. For instance, vHeat-T achieves a throughput of 1514 images/s, $87\%$ higher than Vim-S, $26\%$ higher than ConvNeXt-T, and $22\%$ higher than Swin-T, while maintaining a performance superiority, respectively.

Table 1: Performance comparison of image classification on ImageNet-1K. Test throughput values are measured with an A100 GPU, using the toolkit released by (Wightman, 2019), following the protocol proposed in (Liu et al., 2021). The batch size is set as 128, and the PyTorch version is 2.2.

| Method | Image size | #Param. | FLOPs | Test Throughput (img/s) | ImageNet top-1 acc. (%) |
|---|---|---|---|---|---|
| Swin-T (Liu et al., 2021) | $224^2$ | 28M | 4.6G | 1242 | 81.3 |
| ConvNeXt-T (Liu et al., 2022b) | $224^2$ | 29M | **4.5G** | 1198 | 82.1 |
| DCFormer-SW-T (Li et al., 2023) | $512^2$ | 28M | **4.5G** | - | 82.1 |
| Vim-S (Zhu et al., 2024) | $224^2$ | **26M** | 5.3G | 811 | 81.4 |
| vHeat-T (Ours) | $224^2$ | 29M | 4.6G | **1514** | **82.2** |
| Swin-S (Liu et al., 2021) | $224^2$ | 50M | 8.7G | 720 | 83.0 |
| ConvNeXt-S (Liu et al., 2022b) | $224^2$ | 50M | 8.7G | 687 | 83.1 |
| DCFormer-SW-S (Li et al., 2023) | $512^2$ | 50M | 8.7G | - | 82.9 |
| vHeat-S (Ours) | $224^2$ | 50M | **8.5G** | **945** | **83.6** |
| Swin-B (Liu et al., 2021) | $224^2$ | 88M | 15.4G | 456 | 83.5 |
| ConvNeXt-B (Liu et al., 2022b) | $224^2$ | 89M | 15.4G | 439 | 83.8 |
| RepLKNet-31B (Ding et al., 2022b) | $224^2$ | 79M | 15.3G | - | 83.5 |
| DCFormer-SW-B (Li et al., 2023) | $512^2$ | 88M | 15.4G | - | 83.5 |
| Vim-B (Zhu et al., 2024) | $224^2$ | 98M | 19.0G | 294 | 83.2 |
| vHeat-B (Ours) | $224^2$ | **68M** | **11.2G** | **661** | **84.0** |

**Object Detection and Instance Segmentation.** As a backbone network, vHeat is tested on the MS COCO 2017 dataset (Lin et al., 2014) for object detection and instance segmentation. We load classification pre-trained vHeat weights for downstream evaluation. Considering the input image size is different from the classification task, the shape of FVEs or $k$ should be aligned to the target image size on downstream tasks. Please refer to Sec. D.1 for ablation of interpolation for downstream tasks. The results for object detection are summarized in Table 2, and vHeat enjoys superiority in box/mask Average Precision ($AP^b$ and $AP^m$) in both of the training schedules (12 or 36 epochs). For example, with a 12-epoch fine-tuning schedule, vHeat-T/S/B models achieve object detection mAPs of $45.1\%/46.8\%/47.7\%$, outperforming Swin-T/S/B by $2.4\%/2.0\%/0.8\%$ mAP, and ConvNeXt-T/S/B by $0.9\%/1.4\%/0.7\%$ mAP, respectively. With the same configuration, vHeat-T/S/B achieve instance segmentation mAPs of $41.2\%/42.3\%/43.0\%$, outperforming Swin-T/S/B and ConvNeXt-T/S/B. The

advantages of vHeat persist under the 36-epoch (3×) fine-tuning schedule with multi-scale training. Besides, vHeat enjoys much higher inference speed (FPS) compared with Swin and ConvNeXt. For example, vHeat-B achieves **20.2** images/s, **46%/43%** higher than Swin-B/ConvNeXt-B (13.8/14.1 images/s). These results highlight vHeat's potential to deliver strong performance and efficiency in dense prediction downstream tasks.

**Semantic Segmentation.** The results on ADE20K are summarized in Table 2 (right), and vHeat consistently achieves superior performance. For example, vHeat-B respectively outperform NAT-B (Hassani et al., 2023) and ViL-B (Alkin et al., 2024) by 1.1%/0.8% mIoU.

Table 2: **Left**: Results of object detection and instance segmentation on COCO dataset. FLOPs are calculated with input size $1280 \times 800$. $AP^b$ and $AP^m$ denote box AP and mask AP, respectively. The notation '1×' indicates models fine-tuned for 12 epochs, while '3×MS' denotes the utilization of multi-scale training for 36 epochs. **Right**: Results of semantic segmentation on ADE20K using UperNet (Xiao et al., 2018). FLOPs are calculated with the input size of $512 \times 512$.

| Mask R-CNN 1× schedule on COCO | | | | |
|---|---|---|---|---|
| Backbone | $AP^b$ | $AP^m$ | FPS (images/s) | FLOPs |
| Swin-T | 42.7 | 39.3 | 26.3 | 267G |
| ConvNeXt-T | 44.2 | 40.1 | 29.3 | **262G** |
| vHeat-T (Ours) | **45.1** | **41.2** | **32.7** | 272G |
| Swin-S | 44.8 | 40.9 | 19.7 | 359G |
| ConvNeXt-S | 45.4 | 41.8 | 20.2 | 349G |
| vHeat-S (Ours) | **46.8** | **42.3** | **25.9** | **348G** |
| Swin-B | 46.9 | 42.3 | 13.8 | 504G |
| ConvNeXt-B | 47.0 | 42.7 | 14.1 | 486G |
| vHeat-B (Ours) | **47.7** | **43.0** | **20.2** | **432G** |
| Mask R-CNN 3× MS schedule on COCO | | | | |
| Swin-T | 46.0 | 41.6 | 26.3 | 267G |
| ConvNeXt-T | 46.2 | 41.7 | 29.3 | **262G** |
| vHeat-T (Ours) | **47.2** | **42.4** | **32.7** | 272G |
| Swin-S | 48.2 | 43.2 | 19.7 | 359G |
| ConvNeXt-S | 47.9 | 42.9 | 20.2 | 349G |
| vHeat-S (Ours) | **48.8** | **43.7** | **25.9** | **348G** |

| UperNet on ADE20K | | | |
|---|---|---|---|
| Backbone | mIoU | FPS (images/s) | FLOPs |
| Swin-T | 44.4 | 31.8 | 237G |
| ConvNeXt-T | 46.0 | **37.8** | **235G** |
| ViL-S | 46.3 | - | - |
| vHeat-T (Ours) | **46.9** | 36.7 | **235G** |
| Swin-S | 47.6 | 22.1 | 261G |
| NAT-S | 48.0 | 23.1 | **254G** |
| ConvNeXt-S | 48.7 | **27.7** | 257G |
| vHeat-S (Ours) | **49.1** | 26.1 | **254G** |
| Swin-B | 48.1 | 19.2 | 299G |
| NAT-B | 48.5 | 20.8 | **285G** |
| ViL-B | 48.8 | - | - |
| ConvNeXt-B | 49.1 | 21.6 | 293G |
| vHeat-B (Ours) | **49.6** | **23.6** | 293G |

**Robustness evaluation.** To validate the robustness of vHeat, We evaluated vHeat-B on out-of-distribution classification datasets, including ObjectNet (Barbu et al., 2019) and ImageNet-A (Hendrycks et al., 2021). The results are presented in Table 3. We measure the Top-1 accuracy (%) for these two benchmarks. It is evident that vHeat outperforms Swin and ConvNeXt significantly (better results are marked in bold). These experiments highlight vHeat's robustness across out-of-distribution data, such as rotated objects, different view angles (ObjectNet), and natural adversarial examples (ImageNet-A).

Table 3: Robust comparison of vHeat-B with Swin-B.

| Model | ObjectNet top-1 acc. (%) | ImageNet-A top-1 acc. (%) |
|---|---|---|
| Swin-B | 25.4 | 36.0 |
| ConvNeXt-B | 26.1 | 36.5 |
| vHeat-B (Ours) | **26.7** | **36.8** |

**Low-level vision tasks.** To validate the generalization of our proposed vHeat, we replaced self-attention modules with HCOs in SwinIR (Liang et al., 2021) to form vHeatIR, and tested its performance on several low-level vision tasks with the same settings with SwinIR, including grayscale/color image denoising on Set12 (Roth & Black, 2005)/McMaster (Zhang et al., 2011) and JPEG compression artifact reduction on LIVE1 (Sheikh, 2005). The results are summarized in Table 4, and vHeatIR achieves outstanding results compared to other baseline models, which may be attributed to HCO's computation in the frequency domain. After training for a short period (15000 iterations), the visualization of color image denoising ($\sigma = 15$) is shown in Fig. 5, where vHeatIR outputs

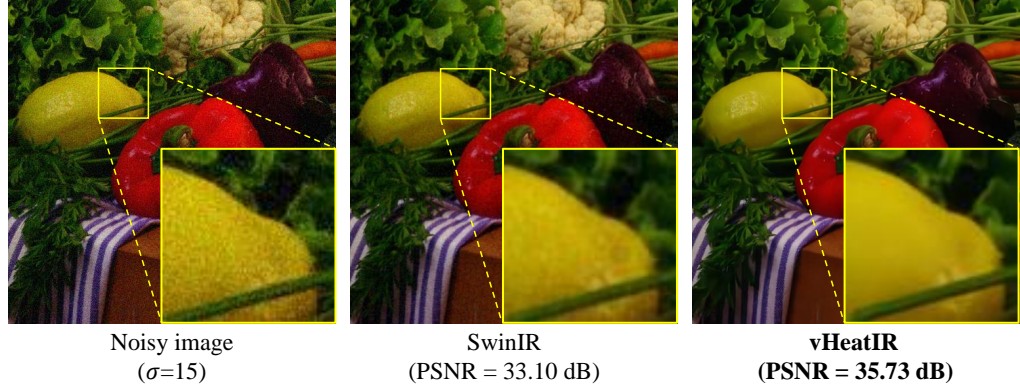

| Noisy image | SwinIR | **vHeatIR** |
|:---:|:---:|:---:|
| ($\sigma$=15) | (PSNR = 33.10 dB) | **(PSNR = 35.73 dB)** |

Figure 5: Color image denoising visualization of vHeatIR and SwinIR after 15000 training iterations ($\sigma = 15$). The input image is selected from McMaster (Zhang et al., 2011).

Table 4: Quantitative comparison (average PSNR) on low-level vision tasks. [†] Results are reproduced for a fair comparison.

| Model | Grayscale/Color Image Denoising (Set12/McMaster, $\sigma = 15$) | JPEG Compression Artifact Reduction (LIVE1, $q = 40$) |
|---|:---:|:---:|
| DnCNN (Zhang et al., 2017) | 32.86/33.45 | 33.96 |
| DRUNet (Zhang et al., 2022) | 33.25/35.40 | 34.58 |
| SwinIR[†] (Liang et al., 2021) | 33.33/35.55 | 34.61 |
| vHeatIR (Ours) | **33.37/35.60** | **34.64** |

a much cleaner image than SwinIR. The experimental results have validated the potential and the generalization on low-level vision tasks of vHeat.

**Computational cost.** The comparisons of throughput / GPU memory / FLOPs of vHeat-B and other ViTs are shown in Fig. 6. Thanks to HCO's $\mathcal{O}(N^{1.5})$ computational complexity *w.r.t.* $N$ image patches, vHeat-B has a significant superiority over other base-level ViT models on throughput/FLOPs. Fig. 6 (Right) shows that with the increase of input image resolution, vHeat enjoys the slowest increase of computational overhead. Fig. 6 (Mid) shows that vHeat requires 80% GPU memory less than Swin-Transformer given large input images. Given the larger image resolution, the superiority becomes larger. These demonstrate vHeat's great potential to handle high-resolution images.

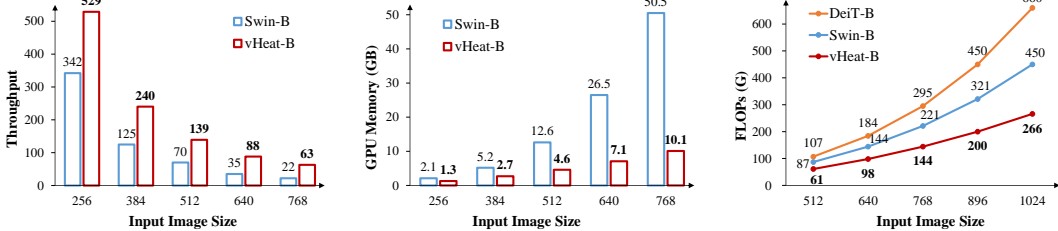

Figure 6: **Left / Mid / Right**: Throughput / GPU memory / FLOPs under different image resolutions. The throughput and GPU memory are tested on 80 GB Tesla A100 GPUs with batch size 64. Swin-B is tested with scaled window size here.

### 4.2 ANALYSIS OF DYNAMIC LOCALITY

**Visual Heat Conduction.** The proposed vHeat works upon an adaptive filtering mechanism. To verify this claim, in Fig. 7, we visualize the temperature $U^t$ defined in equation 10 under predicted $k$ when a random patch is taken as the heat source. With a predicted $k$, vHeat delivers self-adaptive visual heat conduction. As the heat conduction time ($t$) increases, the correlation between the selected

patch and the entire image improves, which effectively filters out unrelated patches in the frequency domain. Please refer to Sec. E in Appendix for vHeat's effective receptive field visualization.

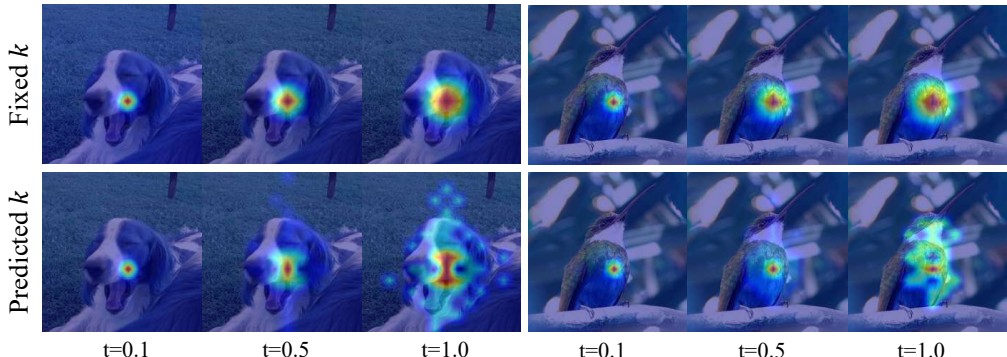

Figure 7: Temperature distribution ($U^t$) when using a randomly selected patch as the heat source. (Best viewed in color)

**Ablation of thermal diffusivity.** To show the effectiveness of shared FVEs, we conduct the following experiments on ImageNet-1K. (1) Fix the thermal diffusivity $k = 0.0/1.0/10.0$. (2) Treat $k$ as a learnable parameter for each layer. (3) Use individual FVEs to predict $k$ for each layer. As shown in Table 5 (Left), when $k = 0.0$, the visual heat conduction doesn't work. A larger fixed $k$ value, e.g., $k = 5.0$, enables HCO to work isotropically without considering the image content and the performance reaches 81.7% top-1 accuracy. Predicting $k$ by FVEs outperforms treating $k$ as a learnable parameter, which may be attributed to the strengthened prior knowledge of frequency values provided by FVEs. Please refer to Sec. D.4 in Appendix for the detailed analysis. When $k$ is predicted by shared FVEs, the performance improves to 82.2%, which validates shared FVEs can effectively reduce the learning diffusivity and further improve the performance.

Table 5: **Left**: Evaluating thermal diffusivity $k$ with vHeat-T. **Right**: Comparison of vHeat with global filters, where vHeat-B$^\star$ denotes replacing HCOs in vHeat-B with operators proposed in GFNet.

| Settings | top-1 acc. (%) |
|---|---|
| Fixed $k = 0.0$ | 81.0 |
| Fixed $k = 1.0$ | 81.7 |
| Fixed $k = 5.0$ | 81.8 |
| $k$ as a learnable parameter | 81.5 |
| Predicting $k$ using individual FVEs | 82.0 |
| Predicting $k$ using shared FVEs | **82.2** |

| Model | #Param. | FLOPs | top-1 acc. (%) |
|---|---|---|---|
| GFNet-H-B | 54M | 8.4G | 82.9 |
| vHeat-S | 50M | 8.5G | **83.6** |
| vHeat-B$^\star$ | 68M | 11.2G | 83.5 |
| vHeat-B | 68M | 11.2G | **84.0** |

### 4.3 COMPARISON WITH GLOBAL FILTERS

To systematically simulate the physical heat conduction, we designed the HCO. Nevertheless, the HCO operates in the frequency domain in practice. Therefore, we compare HCO with (1) GFNet (Rao et al., 2021) (a vision representation model based on global filters in the frequency domain), and (2) replacing HCO with the operators proposed in GFNet for ablation. Results are summarized in Table 5 (Right), vHeat-S has a large superiority over GFNet-H-B under approximate model scale. Besides, replacing HCO with operations proposed in GFNet achieves lower performance, which validates the effectiveness of the proposed HCO and visual heat conduction modeling for representation.

## 5 CONCLUSION

We introduce vHeat, a visual representation model that combines the benefits of global receptive fields, computational efficiency, and enhanced interpretability. The effectiveness of the vHeat model family, including vHeat-T/S/B models, has been demonstrated through extensive experiments and ablation studies, significantly outperforming popular CNNs and ViTs. The results highlight the potential of vHeat as a new paradigm for vision representation learning, offering fresh insights for the development of physics-inspired vision models.

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

## A MOTIVATION

Modern visual representation models are built upon the attention mechanism inspired by biological vision systems. One drawback of it is the lack of a clear definition of the relationship between biological electrical signals and brain activity (energy). This drives us to break through the attention mechanism and attempt other physical laws. Heat conduction is a physical phenomenon in nature, characterized by the propagation of energy. The heat conduction process combines implicit attention computation with energy computation and has the potential to be a new mechanism for visual representation models.

## B HCO IMPLEMENTATION USING $\mathbf{DCT_{2D}}$ AND $\mathbf{IDCT_{2D}}$

Assume a matrix denoted as $\mathbf{A}$ and the transformed matrix denoted as $\mathbf{B}$, the $\mathbf{DCT_{2D}}$ and the $\mathbf{IDCT_{2D}}$ can be performed by

$$\mathbf{DCT_{2D}} : \mathbf{B}_{pq} = \alpha_{\mathbf{p}}\alpha_{\mathbf{q}} \sum_{m=0}^{M-1}\sum_{n=0}^{N-1} \mathbf{A}_{mn}\cos\frac{(2m+1)p\pi}{2M}\cos\frac{(2n+1)q\pi}{2N},$$
$$\mathbf{IDCT_{2D}} : \mathbf{A}_{mn} = \sum_{m=0}^{M-1}\sum_{n=0}^{N-1} \alpha_{\mathbf{p}}\alpha_{\mathbf{q}}\mathbf{B}_{pq}\cos\frac{(2m+1)p\pi}{2M}\cos\frac{(2n+1)q\pi}{2N},$$

(11)

where $0\leq\{p,m\}\leq M-1$, $0\leq\{q,n\}\leq N-1$, $\alpha_{\mathbf{p}} = \begin{cases}\frac{1}{\sqrt{M}}, p=0 \\ \frac{2}{\sqrt{M}}, p>0\end{cases}$, and $\alpha_{\mathbf{q}} = \begin{cases}\frac{1}{\sqrt{N}}, q=0 \\ \frac{2}{\sqrt{N}}, q>0\end{cases}$. $M$ and $N$ respectively denote the row and column sizes of $\mathbf{A}$. Considering the matrix multiplication is GPU-friendly, we implement the $\mathbf{DCT_{2D}}$ and $\mathbf{IDCT_{2D}}$ in Eq. equation 11 by

$$\mathbf{C} = (\mathbf{C}_{mp})_{M\times M} = \left(\alpha_{\mathbf{p}}\cos\frac{(2m+1)p\pi}{2M}\right)_{M\times M},$$
$$\mathbf{D} = (\mathbf{D}_{nq})_{N\times N} = \left(\alpha_{\mathbf{q}}\cos\frac{(2n+1)q\pi}{2N}\right)_{N\times N},$$

(12)

$$\mathbf{B} = \mathbf{CAD^T},$$
$$\mathbf{A} = \mathbf{C^TBD}.$$

Suppose the number of total patches is $N$ and the image is square, the shapes of $\mathbf{A}$, $\mathbf{B}$, $\mathbf{C}$ and $\mathbf{D}$ are all $\sqrt{N}\times\sqrt{N}$, which illustrates the computational complexity of equation 12 and HCO is $O(N^{1.5})$.

We compared our implementation of DCT/IDCT in vHeat with Torch-DCT, which is implemented based on $torch.fft$. Our implemented vHeat-B (661 img/s) is much faster than Torch-DCT (367 img/s), validating our implemented GPU-friendly matrix multiplication is significantly efficient.

## C  EXPERIMENTAL SETTINGS

**Model configurations.** The configurations of vHeat-T/S/B models are shown in Table 6. The FLOPs and training parameters are reported after reparameterization in HCOs.

Table 6: Configurations of vHeat. The contents in the tuples represent configurations for four stages.

| Size | Tiny | Small | Base |
|---|---|---|---|
| Stem | 3×3 conv with stride 2; Norm; GELU; 3×3 conv with stride 2; Norm | | |
| Downsampling | 3×3 conv with stride 2; Norm | | |
| MLP ratio | 4 | | |
| Classifier head | Global average pooling, Norm, MLP | | |
| Layers | (2,2,6,2) (classification) (2,2,5,2) (others) | (2,2,18,2) (classification) (2,2,16,2) (others) | (2,2,18,2) (segmentation) (4,4,20,4) (others) |
| Channels | (96,192,384,768) | (96,192,384,768) | (128,256,512,1024) (segmentation) (96,192,384,768) (others) |

**Image Classification.** Following the standard evaluation protocol used in (Liu et al., 2022a), all vHeat series are trained from scratch for 300 epochs and warmed up for the first 20 epochs. We utilize the AdamW optimizer (Loshchilov & Hutter, 2017) during the training process with betas set to $(0.9, 0.999)$, a momentum of $0.9$, a cosine decay learning rate scheduler, an initial learning rate of $2 \times 10^{-3}$, a weight decay of $0.08$, and a batch size of $2048$. The drop path rates are set to $0.1/0.3/0.5$ for vHeat-T/S/B, respectively. Other techniques such as label smoothing $(0.1)$ and exponential moving average (EMA) are also applied. No further training techniques are employed beyond these for a fair comparison. The training of vHeat-T/S/B takes 4.5/7/8.5 minutes per epoch on Tesla 16×V100 GPUs.

**Object Detection.** Following the settings in Swin (Liu et al., 2022a) with the Mask-RCNN detector, we build the vHeat-based detector using the MMDetection library (Chen et al., 2019). The AdamW optimizer (Loshchilov & Hutter, 2017) with a batch size of 16 is used to train the detector. The initial learning rate is set to $1 \times 10^{-4}$ and is reduced by a factor of $10\times$ at the 9th and 11th epoch. The fine-tune process takes 12 $(1\times)$ or 36 $(3\times)$ epochs. We employ the multi-scale training and random flip technique, which aligns with the established practices for object detection evaluations.

**Semantic Segmentation.** Following the setting of Swin Transfomer (Liu et al., 2021), we construct a UperHead (Xiao et al., 2018) on top of the pre-trained vHeat model to test its capability for semantic segmentation. The AdamW optimizer (Loshchilov & Hutter, 2017) is employed and the learning rate is set to $6 \times 10^{-5}$ with a batch size of 16. The fine-tuning process takes a total of standard $160k$ iterations and the default input resolution is $512 \times 512$.

## D  ADDITIONAL ABLATION STUDIES

### D.1  INTERPOLATION OF FVEs/$k$ FOR DOWNSTREAM TASKS

We have tried several approaches to align the shape for ablation. (1) Directly interpolate FVEs to the target shape of the input image. (2) Add 0 to the lower right region of FVEs to align the target shape. (3) Add 0 to the lower right region of FVEs to $512 \times 512$, and interpolate to the target shape. (4) Directly interpolate the predicted thermal diffusivity $k$ to the target shape. The results are summarized in Table 7. Through the comparison, we select adding 0, then interpolating FVEs to the target shape for all downstream tasks.

### D.2  PLAIN VHEAT MODEL

We've tested the performance of plain vHeat-B on ImageNet-1K classification. Keeping the same as DeiT-B, plain vHeat-B has 12 HCO layers, 768 embedding channels and the patch size is set to 16. Results are shown in Table 8. The superiority of plain vHeat-B over DeiT-B also validates the effectiveness of vHeat model.

Table 7: Evaluating different methods to align the shape of FVEs/$k$ when loading ImageNet-1K pre-trained vHeat-B weights for detection and segmentation on COCO.

| Method | $AP^b$ | $AP^m$ |
|---|---|---|
| Interpolating FVEs to predict $k$ | 47.4 | 42.9 |
| Adding 0 to FVEs | 47.4 | 42.7 |
| Adding 0, then interpolating FVEs | **47.7** | **43.0** |
| Interpolating the predicted $k$ | 47.2 | 42.7 |

Table 8: Plain vHeat-B vs. DeiT-B on ImageNet-1K with 300 epochs supervised training.

| Model | #Param. | FLOPs | Acc |
|---|---|---|---|
| DeiT-B | 86M | 17.5G | 81.8 |
| Plain vHeat-B | 88M | 16.9G | **82.6** |

## D.3 DEPTH-WISE CONVOLUTION

We conduct experiments to validate the performance improvement from DWConv. We replace depth-wise convolution with layer normalization for vHeat-B. Results are summarized in Table 9, and vHeat-B achieves 83.8% Top-1 accuracy on ImageNet-1K classification, 0.2% lower than with DWConv, which validates the main gains come from the proposed HCO. Besides, when $k$ is fixed as a large value, e.g. $k = 10.0$, replacing DWConv with layer normalization causes a significant performance drop (-0.7% top-1 accuracy). The comparison validates predicting $k$ by FVEs can effectively improve the robustness of vHeat.

Additionally, we train vHeat without DWConv with a different recipe from vHeat with DWConv. The batch size is set as 1024, the initial learning rate is set as $1 \times 10^{-3}$, and the weight decay is set as 0.05.

Table 9: Ablation experiments of depth-wise convolution (DWConv).

| Model | DWConv | Acc |
|---|---|---|
| vHeat-B | ✓ | 84.0 |
| vHeat-B | ✗ | 83.8 (-0.2) |
| vHeat-B (fix $k$=10.0) | ✓ | 83.6 |
| vHeat-B (fix $k$=10.0) | ✗ | 82.9 (-0.7) |

## D.4 PREDICTING $k$ BY FVEs $vs.$ TREATING $k$ AS A LEARNABLE PARAMETER

After performing DCT, the features lack explicit frequency value, while FVEs provide the model with prior knowledge of frequency values. Similar to how the introduction of positional encoding can enhance performance even in models that include positional information (Gehring et al., 2017), predicting $k$ by FVEs, rather than treating $k$ as a learnable parameter, reinforces prior frequency information and more clearly represents the relationship between frequency and thermal diffusivity.

## E  RECEPTIVE FIELD VISUALIZATION

The Effective Receptive Field (ERF) (Luo et al., 2016) of an output unit denotes the region of input that contains elements with a non-negligible influence on that unit. In Fig. 8, ResNet, ConNeXT, and Swin have local ERF. DeiT (Touvron et al., 2021) and vHeat exhibit global ERFs. The difference lies in that DeiT has a $\mathcal{O}(N^2)$ complexity while vHeat enjoys $\mathcal{O}(N^{1.5})$ complexity.

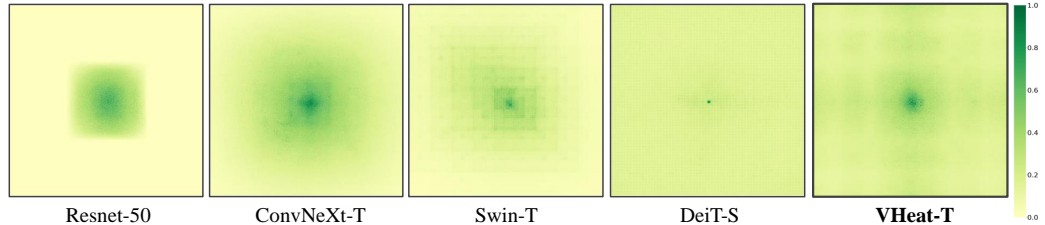

Figure 8: Visualization of the effective receptive fields (ERF) (Luo et al., 2016). The visualization of baseline models are provided from VMamba (Liu et al., 2024). Pixels of higher intensity indicate larger responses with the central pixel.

## F   HEAT CONDUCTION VISUALIZATION

We visualize more instances of visual heat conduction, given a randomly selected patch as the heat source, Fig. 9, validating the self-adaptive visual heat conduction pattern through the prediction of $k$.

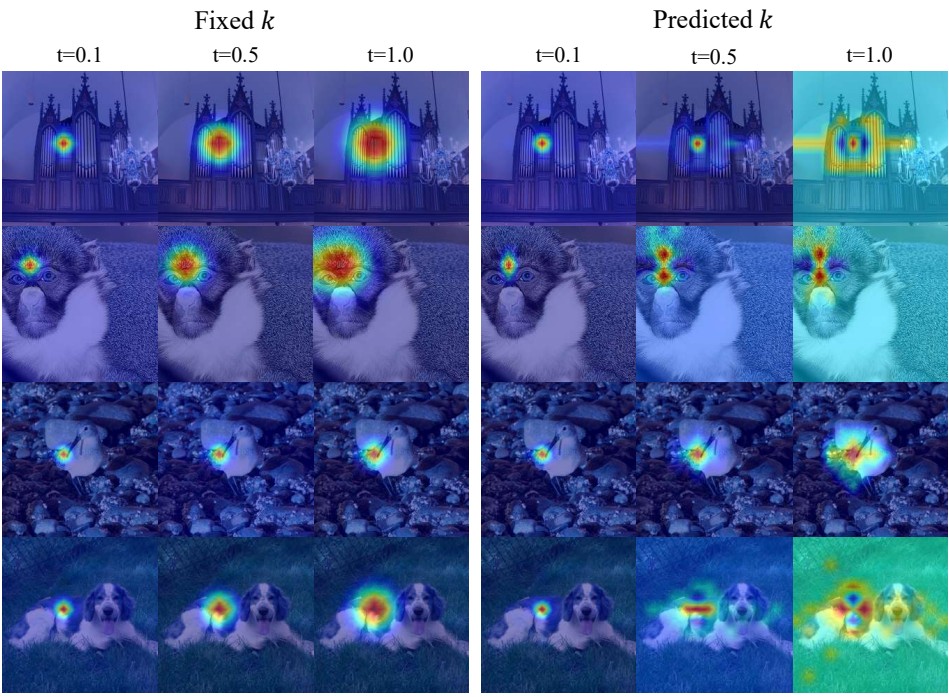

Figure 9: Temperature distribution ($U^t$) when using a randomly selected patch as the heat source. (Best viewed in color)

## G   ANALYSIS OF $k$ IN EACH LAYER

We calculate average values of $k$ in each layer of ImageNet-1K classification pre-trained vHeat-Tiny, Fig. 10. In stage 2 and stage 3, average values of $k$ corresponding to deeper layers are larger, indicating that the visual heat conduction effect of deeper layers is stronger, leading to faster and farther overall content propagation.

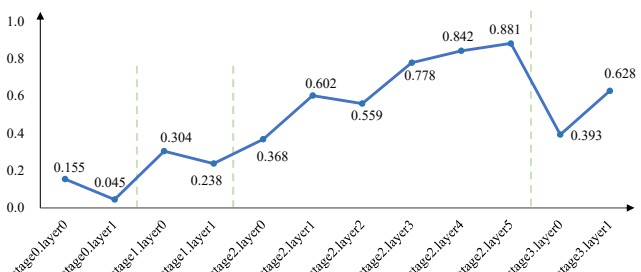

Figure 10: Average values of $k$ in each layer.

## H  FEATURE MAP VISUALIZATION

We visualize the feature before/after HCO in a random layer in stage 2 with randomly selected images as input, Fig. 11. Before HCO, only a few regions of the foreground object are activated. After HCO, almost the entire foreground object is activated intensively.

| Input image | Before the HCO | After the HCO |

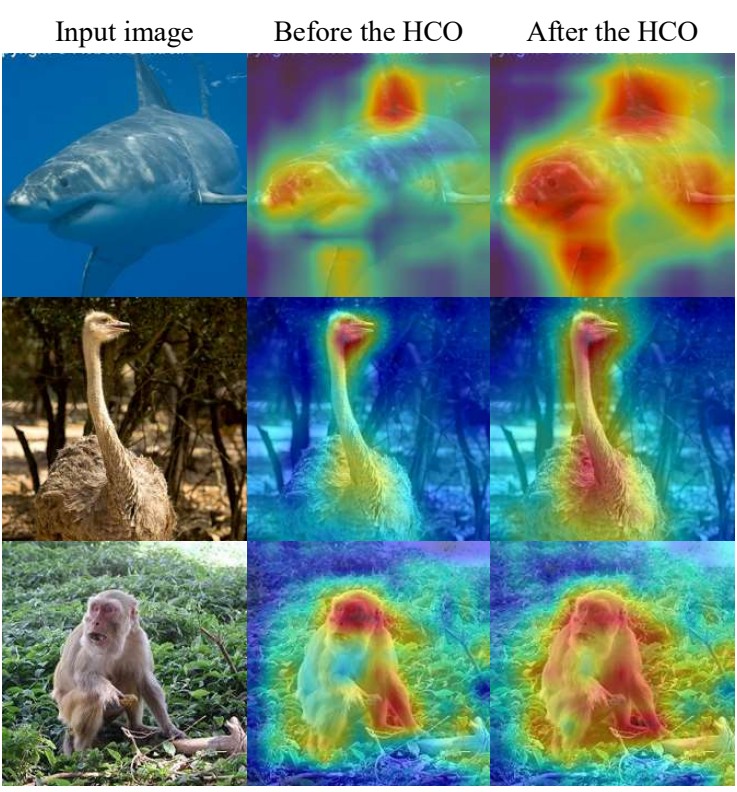

Figure 11: Visualization of the feature before/after HCO in a random layer in stage 2 with ImageNet-1K classification pre-trained vHeat-B. The images are randomly selected from ImageNet-1K.

