# OpenReview forum: "Building Vision Models upon Heat Conduction"
_ICLR.cc/2025/Conference — ICLR 2025 Conference Withdrawn Submission_

### Official Review · Reviewer_yTfo · 2024-11-02

**Soundness:** 2
**Presentation:** 3
**Contribution:** 2
**Rating:** 5
**Confidence:** 4

**Summary:**

In this work, inspired by heat conduction, the authors propose a new method for visual representation called vHeat. vHeat treats each image patch as a source of heat that propagates from local to global, functioning as a bridge between convolution and attention mechanisms. Experimental results suggest that this approach leads to improved performance.

**Strengths:**

1. A novel view of visual representation
2. Extensive experiments and good results.

**Weaknesses:**

1. I would like the authors to discuss the proposed method in relation to relative positional embeddings, as I believe they may share similar high-level concepts.

2. I would also like to understand the motivation behind the additional branch (Linear + SiLU) shown in Fig. 2(b).

3. I am curious about the introduction of DWConv in the block and how much it contributes to the overall performance. While the authors presented this in the supplementary materials, I still do not fully grasp the motivation for its inclusion. Additionally, I would like to know why a different training recipe is adopted (Line 892). Does this lead to an unfair comparison? I was expecting a larger performance gap.

4. While the proposed method achieves promising results, the baselines appear to be somewhat weak and outdated. For example, Table 5 in MetaFormer v2 [1] shows that vHeat performs significantly worse than UniFormer, iFormer, CAFormer, and others if necessary.

5. It’s acceptable if we are not targeting SOTA performance (as well as for downstream tasks), but what other advantages or applications does this method offer that previous work cannot achieve?

**Questions:**

I would expect more discussions about W4 and W5.

---

### Official Review · Reviewer_DFvP · 2024-11-03

**Soundness:** 2
**Presentation:** 3
**Contribution:** 2
**Rating:** 5
**Confidence:** 4

**Summary:**

The manuscript introduces an approach that links spatial propagation of visual semantics with physical heat conduction. It conceptualizes image feature as heat sources and leverages heat conduction principles to extract image features by predicting heat transfer rates. A visual heat conduction operator (HCO) is proposed, resulting in a visual representation model called vHeat, which combines low complexity, global receptive fields, and physical interpretability. Experiments demonstrate the strong performance of vHeat across various visual tasks.

**Strengths:**

1. vHeat’s foundation in the physical principles of heat conduction offers a unique perspective on visual representation. By conceptualizing image patches as heat sources and modeling their interactions through thermal energy diffusion, it provides a fresh, interpretable approach to feature extraction that stands apart from traditional self-attention mechanisms.
2.Despite the lower complexity, vHeat still achieves global receptive fields, allowing it to capture long-range dependencies within images without the computational burden of standard attention mechanisms. This combination of global awareness and efficiency is a strong advantage in many visual tasks.
3.vHeat demonstrates impressive performance across a range of visual tasks, including image classification, object detection, and semantic segmentation. It consistently outperforms existing models like Swin Transformer and ConvNeXt in both accuracy and throughput, highlighting its effectiveness and versatility as a visual representation model.

**Weaknesses:**

1. Although vHeat performs well in experimental settings, the manuscript lacks substantial evidence of its effectiveness in real-world industrial applications. Validation in practical deployments, especially compared to established self-attention models, is necessary to confirm its utility beyond controlled environments.
2.While vHeat draws an analogy between heat conduction and feature propagation, certain aspects of this analogy could benefit from further elaboration. For example, how the thermal diffusivity coefficient 𝑘 corresponds to “feature similarity” or “semantic consistency” in images is not fully explained, which could limit the model’s interpretability.

**Questions:**

1. In physics, heat conduction is defined as the diffusion of energy within a material. Representing image features usually involves pixel or feature map propagation, and whether heat conduction can directly simulate the propagation of visual semantics warrants further discussion. For instance, how the thermal diffusivity coefficient 𝑘 corresponds to "feature similarity" or "semantic consistency" in images requires more specific explanation. The authors could elaborate on how this analogy improves the representation of visual features and explore the biological or physical foundation of this mapping to enhance the method's credibility.
2. While vHeat has improved computational efficiency, its performance in real-world industrial applications has not yet been fully validated. In particular, whether it can outperform existing self-attention models in practical deployments requires further experimental data to substantiate its effectiveness.
3. The manuscript could benefit from a more comprehensive comparison with models inspired by physics and biology, such as diffusion models. Expanding on relevant work in related fields would better highlight the novelty and positioning of this model.
4. Although vHeat's experimental performance is demonstrated, the paper lacks a detailed description of specific parameter settings and model hyperparameters. For instance, how is the thermal diffusivity coefficient 𝑘 adjusted across different levels or data resolutions? How do the learned frequency value embeddings (FVEs) adapt to image content? These settings significantly impact experimental outcomes, and a detailed account of the parameter tuning process and experimental setup is recommended to enhance reproducibility. Additionally, visualizing and comparing heat diffusion effects across different model layers could aid in understanding its feature extraction capabilities.
5. The manuscript primarily focuses on test data in laboratory settings and lacks experimentation in real-world applications. This may limit the model's practical applicability. Including case studies from various fields would be beneficial; if the model demonstrates stable performance in these applications, it would be more persuasive.
6. Although the introduction of physical heat conduction concepts provides a degree of interpretability to the model's computation, the explanation of how specific visual features are enhanced or attenuated through heat diffusion remains insufficient. For example, how certain frequency components of image features are retained or filtered could be clarified. Experimental visualizations showing the influence of heat diffusion on feature selection would be helpful. This improvement in interpretability could aid users in understanding the model's mechanism and applicability, making it more transparent and interpretable.

---

### Official Review · Reviewer_6vYx · 2024-11-03

**Soundness:** 3
**Presentation:** 3
**Contribution:** 3
**Rating:** 6
**Confidence:** 3

**Summary:**

This paper introduces the Heat Conduction Operator (HCO) built upon the physical heat conduction principle, to obtain large receptive fields with less computational overhead.
Its computational complexity is O(N1.5) and is a plug-and-play module to obtain global receptive fields.
Experiments across vision tasks demonstrate its effectiveness.

**Strengths:**

1. The motivation is reasonable.
   2. The result is better than Swin.
   3. Compared with Transformer of O(N2) computational complexity, this method becomes O(N1.5).

**Weaknesses:**

1. The main purpose is to obtain the global receptive fields, thus, some MLP-based Backbones [1-6] should be added to compare.
Maybe, you could discuss computational efficiency and performance in ImageNet, and you can also visualize their receptive fields.

   [1] Strip-MLP: Efficient Token Interaction for Vision MLP. ICCV 2023
   [2] RaMLP: Vision MLP via Region-aware Mixing. IJCAI 2023
   [3] Vision Permutator: A Permutable MLP-Like Architecture for Visual Recognition. TPAMI 2023.
   [4] ResMLP: Feedforward Networks for Image Classification With Data-Efficient Training. TPAMI 2023.
   [5] DynaMixer: A Vision MLP Architecture with Dynamic Mixing. ICML 2022
   [6] MLP-Mixer: An all-MLP Architecture for Vision. NeurIPS 2021

**Questions:**

1. Is the DCT the only choice? Could we replace it with FFT? Could you discuss their potential differences in computational efficiency and numerical stability?

---

### Note · Authors · 2024-11-14

I have read and agree with the venue's withdrawal policy on behalf of myself and my co-authors.